# Immune Checkpoint Glycoproteins Have Polymorphism: Are Monoclonal Antibodies Too Specific?

**Mehrsa Jalalizadeh [1], Reza Yadollahvandmiandoab [1] and Leonardo Oliveira Reis [1,2,*]**

[1] UroScience, School of Medical Sciences, University of Campinas, UNICAMP, Campinas 13083-970, SP, Brazil

[2] Center for Life Sciences, Pontifical Catholic University of Campinas, PUC-Campinas, Campinas 13034-685, SP, Brazil

* Correspondence: reisleo@unicamp.br; Tel.: +55-019-35217481

**Abstract:** Since the 2018 Nobel prize in medicine was granted to the discovery of immune escape by cancer cells, billions of dollars have been spent on a new form of cancer immunotherapy called immune checkpoint inhibition (ICI). In this treatment modality, monoclonal antibodies (mAbs) are used to block cell-surface glycoproteins responsible for cancer immune escape. However, only a subset of patients benefit from this treatment. In this commentary, we focus on the polymorphism in the target molecules of these mAbs, namely PD-1, PD-L1 and CTLA4; we explain that using a single mAb from one clone is unlikely to succeed in treating all humans because humans have a genotype and phenotype polymorphism in these molecules. Monoclonal antibodies are highly specific and are capable of recognizing only one epitope ("monospecific"), which makes them ideal for use in laboratory animals because these animals are generationally inbred and genetically identical (isogenic). In humans, however, the encoding genes for PD-1, PD-L1 and CTLA4 have variations (alleles), and the final protein products have phenotype polymorphism. This means that small differences exist in these proteins among individual humans, rendering one mAb too specific to cover all patients. Our suggestion for the next step in advancing this oncotherapy is to focus on methods to tailor the mAb treatment individually for each patient or replace a single clone of mAb with less specific alternatives, e.g., a "cocktail of mAbs", oligoclonal antibodies or recombinant polyclonal antibodies. Fortunately, there are ongoing clinical trials on oligoclonal antibodies at the moment.

**Keywords:** cancer immunotherapy; immune checkpoint receptors; CTLA4; PD-1; PD-L1; therapeutic monoclonal antibody; inhibitors; blockers; polyclonal antibody; oligoclonal antibody; polymorphism; variations; tumor plasticity; antigenic drift

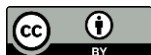

## 1. Introduction

*We are each unique, and so is our immune system.*

In the 1990s, Honjo and his colleagues described a glycoprotein on the surface of T cells that functioned as an immune "brake"; activating the molecule seemed to halt immune responses. Honjo called the molecule Programmed cell Death-1 (PD-1), as it was observed that once activated, it causes apoptosis of the immune cell, a crucial mechanism in negative selection of T cells and prevention of autoimmunity. A few years later, Allison and his colleagues described a similar glycoprotein (CD125) with almost identical properties. Allison renamed the molecule as Cytotoxic T-lymphocyte Antigen 4 (CTLA4) and realized that cancer cells use the molecule to evade the immune system [1,2].

Both of these molecules are now considered as immune checkpoint (IC) receptors and are postulated to function as some form of "brake" on the immune system; when activated, they seem to "stop" the immune cells from attacking through various mechanisms [3]. Both receptors are activated by ligands from the B7 superfamily. Allison created an antibody that blocked the interaction of CTLA4 with its ligand and observed that mice

injected with the antibody were able to eliminate cancer using their immune system. This cancer treatment modality gained attention when a 2010 phase I clinical trial on melanoma patients showed slightly longer overall survival in the treatment arm [4]. Since then, multiple monoclonal antibodies (mAb) have been created and approved by the FDA that block either CTLA4, PD-1 or PD-1's ligand PD-L1. These mAbs are designed to remove the brake on the immune system and reactivate the patient's T cells to attack the tumor cells in a treatment modality called immune checkpoint inhibition (ICI).

Bladder cancer is one of the cancers approved to be treated by ICI, mainly as second-line treatment in advanced diseases. Initially, pembrolizumab and nivolumab as PD-1 inhibitors and atezolizumab, avelumab and durvalumab as PD-L1 inhibitors were approved for metastatic cancer unresponsive to or ineligible for first-line treatment. These mAbs have been tested alone or in combination with CTLA4 blocking mAbs tremelimumab or nivolumab [5].

Unfortunately, these agents proved to be far from perfect as more clinical studies were conducted on them. At best, 30% of patients with metastatic urothelial carcinoma of the bladder were shown to benefit for a few months from ICI, while some of these mAbs failed to show any survival benefit and had worse adverse effects than chemotherapy. Some mAbs were voluntarily withdrawn from the market by the provider without disclosing a clear reason [6,7].

We know that monoclonal antibodies are highly specific [8–10]; biotechnology science uses the term "monospecific" to describe mAbs' affinity to a single epitope and their inability to identify variations of the target. A small change in their target epitope makes them blind to the target. In this commentary, we challenge the rationale behind using mAb from a single clone to target IC receptors in all humans because, unlike inbred laboratory animals, humans have a polymorphism in these molecules. Therefore, a monospecific mAb is unlikely to be a universal treatment for all patients. In the first part of this commentary, we describe the studies that prove polymorphism in the target of ICI mAbs. We explain that extensive evidence on this polymorphism is already available. In the second part of the commentary, we explain possible reasons as to why this polymorphism has been overlooked. In the final part, we discuss possible suggestions to solve the polymorphism issue, namely replacing a single clone of therapeutic mAb with "mAb cocktails", oligoclonal antibodies or newer generations of polyclonal antibodies. These suggestions must be proven clinically effective, and some of them are currently being tested in clinical trials [11]. We hope that targeting the polymorphism issue of ICI targets will improve and extend ICI efficacy for a higher number of individuals.

## 2. Section 1: Evidence on Polymorphism of Immune Checkpoint Receptors

### 2.1. Genotype Polymorphism

In humans, CTLA4 and PD-1 are encoded on the q arm of chromosome 2 (2q33 and 2q37, respectively), and PD-L1 is encoded on the p arm of chromosome 9 [12–14]. Multiple studies have discovered various alleles for their genes [15–18]. A quick search through the gene bank of the National Library of Medicine reveals multiple single-nucleotide polymorphisms (SNPs) in many exon regions of all three genes. For example, the rs2227981 SNP for PD-1 is shown by 30 different citations to have two variations with almost equal frequencies in different ethnicities (1:1 or 1:1.5 frequencies) [19]. Importantly, very few of these genetic polymorphisms have been associated with risk of cancer [15] or autoimmune diseases [18]. This means that most of these SNPs are not pathologic mutations—rather, *normal* variations. Moreover, the chromosomal locations of CTLA4 and PD-1 are perfect for cross-over during miosis; they are located on the long arm of the second largest human chromosome just above telomeres, giving them ample room to match and cross-over during miosis to create new combinations in the population [20].

*2.2. End Product (Phenotype) Polymorphism*

Not all gene alterations end up in the final product. Alterations in introns are sliced out before translation, and even variations in exons can be sliced out of the final protein or involve amino acids that do not change the properties of the protein. However, ample evidence is available on polymorphism in the final immune checkpoint glycoproteins among humans. Most importantly, evidence shows variation in the interaction between mAbs and immune checkpoint glycoproteins of individual patients.

The strongest evidence in favor of our argument is provided by eight separate studies that compared different mAbs assays in terms of their ability to stain PD-L1 in specimens from human patients [21–28]. All eight studies showed imperfect inter-assay correlations. For example, in the study by Eckstein et al., there were 10 patients who only stained positive for PD-L1 using Ventana SP263 mAb clone, while the other mAb clones (DAKO 28-8, DAKO 22c3 and Ventana SP142) failed to stain the specimens of those 10 patients. The studies called this imperfect "inter-assay correlations"; however, we believe that this is strong proof that a single clone of mAb is not sensitive enough to cover all individual polymorphisms between patients. The following two scenarios can further be postulated:

1. When an mAb assay fails to stain a tumor specimen from a patient, it does NOT mean that the antigen is absent; it can mean that the specific epitope for this clone of staining mAb is absent, and the assay is unable to detect the variant that this patient is expressing. One assay of a single clone of mAb is not sensitive enough to rule out the expression of these molecules due to their polymorphism.

2. If a patient specimen is stained positive for the presence of the antigen, it does NOT mean that the treatment mAb will also recognize the antigen. Staining mAbs are designed to attach to formaldehyde fixated glycoproteins at room temperature and are completely different from those designed to be injected and attached to the natural form of the glycoprotein in physiological body state. According to Brown et al. [29], treatment mAbs have a polymorphism in their affinity to different epitopes.

Newer clinical trials stain all their patients' tumor samples before initiating therapy and have shown that the response to ICI is independent of whether the patient was stained positive or negative [6]. This further proves that we should refrain from marking a patient tumor as "not expressing" the receptor if one mAb assay fails to stain their sample.

*2.3. Further Evidence*

The following evidence *speaks in favor of* individual variation in these glycoproteins.

PD-1 and CTLA4 belong to the immunoglobulin (Ig) superfamily, and both contain a variable domain (V domain or V-set domain). Through the process of "DNA rearrangement", the Ig V domain is theoretically capable of forming $2 \times 10^{12}$ varying combinations in a single individual. This means that even in the same human, the PD-1 and the CTLA4 can potentially vary between cell lines. The V domain is found to be crucial in the interaction of both these glycoproteins with their ligands; in particular, CTLA4 uses the V domain for attachment. Any mAb designed to block this attachment may partially or fully cover the V domain as its epitope to attach with a higher affinity than the ligand. Hence, if the V domain is part of the epitope, that mAb may be specific for one cell line in the same human and miss other cell lines. Recently, clinicians have been wondering whether the development of adverse effects during ICI therapy signals the success of treatment. According to this logic, if the mAb epitope overlaps with the V domain of the target, it is likely that the presence of adverse effects does NOT show success in eliminating cancer or even predicts failure [30–32].

Lastly, in 1994, Honjo et al. [13] showed that the human and murine PD-1 only share 60% of their amino acids. Later, Lin et al. [33] showed that the murine PD-1 is able to attach to human PD-L1 despite the 40% difference in structure [33]. This means that PD-1 can theoretically change up to 40% of its protein structure without changing its function. However, this is a very crude judgement.

### 3. Section 2: Why Is This Polymorphism Overlooked?

In the first section of the commentary, we explain that polymorphism in ICI targets is not a new subject, and it has already been extensively studied and proven. The success of mAb in animal studies cannot extend to humans due to the following reasons:

a. Laboratory animals are inbred and highly identical (isogenic or semi-isogenic). Individual differences in the PD-1, PD-L1 and CTLA4 are less expected in generationally inbred animals.

b. Tumors in laboratory animals are often created by injection of a well-established cultured cell line; polymorphism is less expected in these cell cultures.

c. Laboratory animals are kept in sterile conditions. Their immune system is therefore naïve, untrained and can act differently from humans who have been in contact with myriads of pathogenic and non-pathogenic germs since birth.

Furthermore, the human studies of these drugs initially included very few selected subjects. Phase II clinical trials usually involve 25 to 100 patients and do not have a placebo arm. In small sample sizes, statisticians usually try to reduce variation between subjects by selecting similar individuals. None of the initial ICI clinical trials mention the race or ethnicity of their included subjects; however, we can assume they were probably of the same race and similar background. This would reduce the individual variation in their target proteins. This also explains why these treatment modalities have lower chance of success in phase III clinical trials or confirmatory phase III clinical trials with larger sample sizes. The results of the larger trials indicate that mAbs against immune checkpoint receptors seem to "benefit only a *subset* of patients".

### 4. Section 3: Authors' Suggestions

#### 4.1. The Monospecific Nature of mAbs Is a Double-Edged Sword

The "monospecific nature" of mAbs is especially troublesome when the target has polymorphism [9,10]. Evidence provided in this commentary shows that polymorphism on IC receptors is not a new topic and has been extensively studied. However, a single clone of mAb has been seen as a panacea for all humans. In the next section of this article, we describe two possible solutions to address the polymorphism issue in ICI, both of which are unfortunately costly and require extensive funding for research. We strongly urge the funding needs of the following potential treatment solutions to be met, with a USD 25 billion yearly budget prediction [34]; however, we it is hard to halt the treatments until the polymorphism issue is solved.

#### 4.2. First Solution: Reduce Specificity and Increase Sensitivity

4.2.1. Polyclonal Antibodies and Recombinant Polyclonal Antibodies

Polyclonal antibodies (pAb) have been used for decades against poorly characterized targets. Toxins and venoms are perfect targets of pAbs; various snake venoms can be neutralized by a single batch of pAb if the venoms are similar [35]. However, pAbs have always been feared in immunotherapy for being too non-specific and unpredictable. The concentration of the specific antibody in a pAb product is between 50 and 200 μg/mL, while mAbs have a 10-fold higher concentration of the specific antibody. mAbs also have the benefit of being more reproducible from batch to batch; the single B-cell clone that produces the mAb is immortalized, while pAbs are generated by polyclonal B cells that are not immortalized, requiring the creation of a new generation of reacting cells every time the cells die. The other advantage of mAbs is that they are unlikely to cross-link, fix complement or activate the target. The latter advantage is important in ICI therapy, as the antibody must only neutralize the IC receptors on the immune cells [8].

Unfortunately, the multiple benefits of pAbs have been overlooked due to these concerns. A single batch of pAb is capable of recognizing multiple epitopes and covers varieties of the target, rendering pAb assays more sensitive than mAbs. Furthermore, pAb is

less sensitive to challenging environments. For example, mAbs have the problem of losing their efficacy when the temperature or the pH is not ideal [36,37]. Aside from being immune to challenging environments, pAbs have the ability to detect the target both in a fixated state (formaldehyde fixation prior to staining) and in its natural state (in vivo). This is a huge advantage in ICI therapy, allowing the same batch of therapeutic pAb to first be tested on the fixated tissue sample of the patient. mAbs, on the other hand, have to be produced separately for either staining purposes or therapeutic purposes [9].

Recombinant pAbs (r-pAb) are the third generation of antibodies introduced in 1994 to address the problem of the monospecific nature of mAbs while avoiding the common issues of pAbs, including low reproducibility and low concentration of the specific antibody. These antibody products are capable of attacking multiple epitopes and are more potent than mAbs in inducing cytotoxic cellular response [10,38]. The cytotoxic cellular response induction is beneficial in ICI therapy if the target receptor is only expressed on the cancer cells. However, if the target receptor is mainly expressed on the immune cells, both pAbs and r-pAbs should theoretically be avoided.

A possible solution to the low specificity of pAb is taking advantage of the very important aspect of ICI targets. IC receptors are called immune "brakes" for an important reason; they only function when the immune system is "already in action". This characteristic can enhance the specificity of a systemically injected pAb if the immune system is first activated against the tumor using a focal intervention, e.g., focused radiotherapy. We have previously described oncotherapy treatments that have the ability to both kill the target cell and call the immune system against the cell. This is called immunogenic cell death (ICD), and it is induced by various treatments, such as radiotherapy [39]. Radiation therapy is perfect for this treatment combination, as it can be used focally on a limited anatomical region. After the immune system is activated focally against cancer by an ICD inducer, the treatment can be complemented by a systemic pAb to cover the cancer cell's IC receptors and ensure that they are unable to shut down the T-cell response. Theoretically, a polyclonal antibody mixture can have fewer side effects in this combination modality.

The extra advantage of the pAb mixture is its potential to protect against future mutations of its target, i.e., antigenic drift. Antigenic drift was first described for viruses with a high mutation rate, but it is also observed in cancer cells as a mechanism of immune evasion through escaping T-cell recognition [40]. Tumor cell "plasticity" refers to tumor resistance to drugs by phenotype switching without new mutations [41,42]. In the case of ICI resistance, this is possible if the tumor cells switch between two alleles of one IC receptor, e.g., the paternal allele is no longer expressed because its epitope was recognized by the treatment mAb; the new tumor cells now explicitly express the maternal allele. mAbs tend to lose their effectiveness against "highly mutagenic targets" and do not cover plasticity of tumor cells, while a polyclonal mixture of antibodies is less sensitive to these mechanisms of treatment resistance [9,10].

### 4.2.2. Monoclonal "Cocktails" or Oligoclonal Antibodies

Mixing different mAbs against the same receptor is a theoretical alternative to pAb with a more predictable response. This is not a new concept, and multiple companies have produced biclonal or oligoclonal antibodies that have higher sensitivity than mAbs; however, they are poorly studied in oncotherapy [43–45]. Fortunately, new antibody combinations are under development [11]. Ideally, different companies that have produced mAbs against immune checkpoint receptors should combine their treatments and offer the mixture as a potential universal treatment for all patients of all ethnicities. This is slightly challenging, since some epitopes overlap [29], and the ideal cocktail must not include mAbs that interfere with each other.

*4.3. Second Solution: Increase Specificity to the Individual Level and Tailor It*

We suggest refraining from using the term "not expressed" when describing IC receptors in humans. Instead, we propose describing that the "epitope" for the mAb clone X was not detected on sample cells of this patient. Each patient sample should be stained with not just one mAb assay but multiple to choose the highest performing mAb with the ideal epitope for the patient (similar to cross-matching). If companies match the epitope of their staining mAb with the epitope of a treatment mAb, the pathologist can easily record the presence of a specific "epitope" on the cancer cell and recommend the treatment mAb that attaches to the exact same epitope. This is financially challenging, as companies must produce perhaps hundreds of mAb with low sensitivity and high specificity for each variation of the IC receptor. Furthermore, companies must take samples from humans of different ethnic backgrounds to generate their mAb clones.

If tumor resistance is developed under mAb therapy against CLTA4 or PD-1, it could mean that the tumor cells have changed their configuration of PD-1 or CTLA4. As mentioned before, both of these receptors contain the V domain of the immunoglobulin superfamily, which could change configuration through the process of DNA rearrangement. A resistant tumor must be re-tested and re-matched with a new mAb.

Future studies should compare different mAb assays that stain PD-1 and CTLA4. So far, eight studies [21–28] have found imperfect correlation between different mAb clones in detecting PD-L1 on human sample cell surfaces. However, no studies have compared this with PD-1 and CTLA4 antibodies.

Lastly, while roughly 8% of the melanoma patients in the initial clinical trials showed benefit from ICI therapy, the treatments were described as "strikingly effective" on the Nobel prize website [46]. These glib attitudes toward newly discovered treatments should be avoided, as they lead to oversimplification of an extremely complicated biological phenomenon.

**Author Contributions:** M.J.: Literature Review, Hypothesis Formation, Manuscript Writing; R.Y.: Manuscript Editing; L.O.R.: Guidance, Editing. All authors have read and agreed to the published version of the manuscript.

**Funding:** This commentary did not receive any funding.

**Conflicts of Interest:** The authors declare no conflicts of interest.

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
