# Peer review of "Immune Checkpoint Glycoproteins Have Polymorphism: Are Monoclonal Antibodies Too Specific?"

_curroncol, doi:10.3390/curroncol30010098_

Round 1

Reviewer 1 Report

A polymorphism of ICs isn't a new point, there are several studies that have discussed this phenomenon. Furthermore, the low response to ICIs doesn't mean there is no successful interaction between ICIs and the receptor because some studies evidenced the decreased expression of ICs post-treatment without clear immune activation or tumor targeting. Thus, the story isn't only about the complete silencing of ICs in patients. 

On the other hand, the examples of combination authors mentioned weren't matched with what the authors urged. the combination of anti-PD1 and anti-CTLA4 is totally different than polyclonal antibodies for targeting one receptor only. 

In addition, the clinical side effects of polyclonal antibodies are serious and can not be ignored 80% of patients showed serious reactions post-treatment with immunosuppressive polyclonal antibodies and that is the solid reason to withdraw and avoid using these types of drugs.  

Author Response

Reviewer 1

The authors composed a commentary called “Immune Checkpoint Glycoproteins Have Polymorphism, and Monoclonal Antibodies Are Too Specific.” They suggested that mAbs immunotherapy should be designed to match each patient’s needs or be replaced with polyclonal antibodies.

 A polymorphism of ICs isn't a new point, there are several studies that have discussed this phenomenon.

A: Thank you for your comment.

Yes, polymorphism is not a new point, it has already been studied extensively and in this commentary, we gathered all those extensively studied articles.

We are not trying to prove that polymorphism is new, we are trying to show that it is being ignored and it is contributing to treatment failure. We are asking for more attention to this polymorphism and funding to study it. 

We have changed the wording of the introduction of our commentary to reflect our aim more clearly.

Furthermore, the low response to ICIs doesn't mean there is no successful interaction between ICIs and the receptor because some studies evidenced the decreased expression of ICs post-treatment without clear immune activation or tumor targeting. Thus, the story isn't only about the complete silencing of ICs in patients. 

A: Thank you for your comment. We added more explanations on “antigenic drift” and “cancer plasticity” in “Section 3: Authors’ suggestions” line 346.

Staining negative for IC receptors does not necessarily mean that there is decreased expression. It can also mean that the cells have changed the phenotype of their IC receptor, this phenomenon can be due to “antigenic drift” or “cancer plasticity”. In these situations, the patient samples are falsely labeled as “not expressing or low expressing” because of the change in the target epitope, not because the receptor is less expressed. Please refer to the figure (graphic abstract) for better understanding.

On the other hand, the examples of combination authors mentioned weren't matched with what the authors urged. the combination of anti-PD1 and anti-CTLA4 is totally different than polyclonal antibodies for targeting one receptor only. 

A: This review is recently published in MDPI: https://doi.org/10.3390/cancers13184620.

In this review, mixing mAb against two different receptors is mentioned as a possible reason mixing antibodies is better than a single clone of mAb. Yes, they are not the same, this is why immediately before that we mention that we could not find any studies that matched what we urged. Please remember that a “commentary” is supposed to bring light to the LACK of studies.

However, we understand that the sentence can be confusing for readers. We, therefore, removed it from the manuscript.

In addition, the clinical side effects of polyclonal antibodies are serious and can not be ignored 80% of patients showed severe reactions post-treatment with immunosuppressive polyclonal antibodies and that is the solid reason to withdraw and avoid using these types of drugs.  

A: As shown in this review https://doi.org/10.3390/cancers13184620, multiple trials on mixtures of “Oligoclonal antibodies” and “mAb cocktails” are currently underway for the reason that we are explaining in this commentary. Oligoclonal antibodies are something between monoclonal and polyclonal antibodies. We added this to the manuscript in “Section 3: Authors’ suggestion” plus explained other possible alternatives like r-pAbs (recombinant polyclonal antibodies).

Please pay attention that Section 3 brings “suggestions”. Section 1 and 2 are evidence and the final section are suggestions for future studies.

Can reviewer 1 please provide the reference for this statement involving an 80% number? As we suggested in the article, pAb should be used immediately after radiotherapy because radiotherapy is a focal “immunogenic cell death inducer”. As we explained, ICI only works if the immune system is already activated; therefore, activating the immune system using a focal intervention such as radiotherapy can “theoretically” reduce the side effects. In this commentary, we are trying to gather funding to study this combination therapy. Please pay attention to our choice of words; when we say “theoretically”, we use logic, not observations.

Reviewer 2 Report

The authors composed a commentary called “Immune Checkpoint Glycoproteins Have Polymorphism, and Monoclonal Antibodies Are Too Specific.” They suggested that mAbs immunotherapy should be designed to match each patient’s needs or be replaced with polyclonal antibodies.

1.     Authors suggested the combination of pAb with radiotherapy has theoretical advantages, and mixture pAbs has fewer side effects, these are strong statements. Authors need to provide evidence to claim these points, for example, clinical studies or even laboratory scale studies.

2.     Authors may specify the capability of the pAb mixture in neutralizing the receptor.

3.     What is the reference source of the statement that pAb has less tumor-resistant than mAb?

Author Response

Reviewer 2

  1. Authors suggested the combination of pAb with radiotherapy has theoretical advantages, and a mixture pAbs has fewer side effects, these are strong statements. Authors need to provide evidence to claim these points, for example, clinical studies or even laboratory scale studies.

A: Thank you for the comment. We adjusted the manuscript to address this issue.

In this commentary, we are trying to bring the spotlight to the polymorphism of IC receptors and ask for more attention to this problem. The solutions provided in the final section titled “Authors’ suggestions” are not the main point of the commentary and are mere suggestions.

To clear our aim in the manuscript, we divided the manuscript into 3 sections, the 3rd section is titled “Author’s suggestions” and includes this possible combination therapy as a mere suggestion and topic for future studies.

  1. Authors may specify the capability of the pAb mixture in neutralizing the receptor.

 A: Thank you. The pAb suggestion was the secondary endpoint of the commentary. However, we have added multiple references and new insights into pAb characteristics, we further described recombinant pAbs and “oligoclonal” Abs as other possible solutions. We also found a review article https://doi.org/10.3390/cancers13184620 that claims new Ab mixtures are currently under development.

  1. What is the reference source of the statement that pAb has less tumor-resistant than mAb?

A: We added references. We have rewritten and added more evidence to Section 3.

Round 2

Reviewer 1 Report

The content sounds good. The authors improved the manuscript adequately.  

Author Response

Thank you for the opportunity to improve our manuscript.

Reviewer 2 Report

The manuscript is accepted for publishing. 

Author Response

(The authors gave the same response as above.)
